# Unbalance Response Analysis of a Spindle Supported on Gas Bearings: A Comparison between Different Approaches

Federico Colombo *, Luigi Lentini, Terenziano Raparelli, Andrea Trivella and Vladimir Viktorov

Department of Mechanical and Aerospace Engineering, Politecnico di Torino, 10129 Turin, Italy; luigi.lentini@polito.it (L.L.); terenziano.raparelli@polito.it (T.R.); andrea.trivella@polito.it (A.T.); vladimir.viktorov@formerfaculty.polito.it (V.V.)
* Correspondence: federico.colombo@polito.it

**Abstract:** Gas journal bearings are widely employed in high-speed spindles for the micromachining industry. Compared to their oil and rolling counterparts, gas bearings have a longer life span, lower friction and a lower level of noise. In order to design accurate high-speed spindles supported by externally pressurized gas bearings, it is vital to analyze the characteristics of rotor bearing systems. In this paper, we present an analysis of the unbalance response of a high-speed spindle supported by gas journal bearings. A number of aspects to enhance the accuracy of the system are discussed. We performed the analysis by considering a nonlinear and a linearized numerical model validated through experimental measurements.

**Keywords:** hybrid gas bearings; rotordynamics; electro-spindle; unbalance response

## 1. Introduction

The miniaturization of information technology devices such as digital cameras, personal computers and mobile phones has rapidly progressed. The production of these devices has increased the demand for machine tools that produce precision small parts using end mill cutters with diameters of less than 1 mm. This kind of manufacturing process requires spindles with high accuracy and fast rotational speed. Due to their low friction, low vibration transmissibility and long life, gas journal bearings represent the best type of bearings that can be used in such ultra-precision machining tools. In contrast with ball or roller bearings, gas bearings are characterized by a long life span and are free of the high super-synchronous vibrations that compromise precision and lead to high power consumption.

However, gas bearings are not readily available, and each one requires a dedicated design process depending on its particular application. In order to design an accurate high-speed spindle supported by externally pressurized gas bearings, it is vital to analyze the characteristics of rotor bearing systems. This involves the analysis of the static characteristics, making it possible to define a preliminary geometry of the bearing according to the required load capacity and flow rate. Assessment of the dynamic behavior, i.e., dynamic coefficients and stability, is also crucial to guarantee a successful application. An appropriate choice of bearing feeding system is an appropriate starting point. Chen C.-H. et al. [1] elucidated the coupled effects of pressure compensation that is caused by orifice and inherent restrictors in series on the static and dynamic characteristics of a rigid rotor that is supported by orifice-restricted aerostatic bearings. The results of their simulations revealed that the effects of orifice and inherent restrictors in series on bearing performance cannot be ignored in the design of aerostatic bearings. Yoshimoto et al. [2] used both CFD and FEM methods to arrive at the same conclusion and demonstrated the absence of a clear boundary between inherently compensated and orifice restrictors. Otsu et al. [3,4] experimented with the use of surface-restricted layers and compound restrictors. They found that surface-restricted layers generally had the effect of increasing the dynamic performance

of porous journal bearings. Similarly, aerostatic journal bearings with compound restrictors showed higher stiffness and threshold speeds than those obtained with inherently compensated restrictors. Moreover, they showed that the threshold of instability is largely affected by inertia effects when the whirl ratio is greater than two. Park and Kim [5] and Belforte et al. [6,7] proposed new types of feeding systems that could increase the rotor bearing system performance. The integration of external dampers, e.g., O-rings [8,9], air rings [10,11] and bump foil [12,13], makes it possible to increase the dynamic stability of rotor bearing systems. In this regard, Liebich et al. [14] presented a comprehensive analysis on the use of elastomeric rings by proposing a new approach to optimize their material properties. The dynamic performance and stability of journal bearings can be evaluated through nonlinear rotor bearing models, i.e., the orbit method, or models that are obtained by the linearization of the stiffness and damping coefficients of the bearings [8]. The orbit method includes the complete nonlinear equations of the rotor and the bearings, which are numerically integrated to obtain the shaft center orbits corresponding to any set of geometrical, running and initial conditions. This method can be used to calculate the unbalance response, the onset speed and the nonlinear rotor bearing behavior in cases of relatively large displacements. Belforte et al. [15–17] and Liu et al. [18] successfully applied this method in the case of rotors supported by self-acting and floating bearings. Despite their accuracy, nonlinear models can still be time consuming. On the contrary, linearized models are obtained by solving the dynamic Reynolds equations based on the assumption of the small perturbation proposed by Lund [19]. Here, the eight linear stiffness and damping coefficients are obtained to study the stability of the rotor bearing system [20]. Waumans et al. [21] accurately described how to perform this kind of analysis applied to a micro-turbine rotor supported on air bearings by studying its imbalance response and stability.

In this paper, we present the analysis of the unbalance response of a high-speed spindle supported by journal gas bearings. Firstly, we used the nonlinear model of the rotor bearing system that was previously validated in [22] to evaluate the unbalance response. We used these results to quantify the discrepancies with respect to the results obtained through the linearized model. Finally, we employed the linearized model to investigate a number of solutions to improve the spindle accuracy, i.e., to reduce the dynamic runout. We then compared the numerical results with experimental results obtained through cost-down measurements.

## 2. Materials and Methods

The prototype of an electro-spindle supported by hybrid gas journal bearings is illustrated here, see Figure 1. It is designed for drilling and micro-machining applications for rotational speeds up to 100 krpm. The rotor is accelerated by means of a permanent magnet synchronous motor of 3.5 kW output power integrated between the aerostatic journal bearings (JBs). The spindle is cooled by means of a water circuit which interests the stator motor, the JBs and the thrust bearing. The cooling circuit is sealed by means of O-rings positioned between the journal bearings and the external carter. The rotor bearing geometry is specified in Table 1, and Table 2 reports the inertial characteristics of the shaft. Each journal bearing is supplied by means of two series of 12 equally spaced orifices located 7 mm from the edges. The rotor bearing system is sketched in Figure 2, where $l_1$ and $l_2$ refer to the distances of the bearing centers with respect to the rotor center of mass. The right-handed fixed reference system $xyz$ is defined. The rotor radial vibration is measured by means of capacitive sensors (capaNCDT CS05 by Microepsilon, Ortenburg, Germany) placed on two measuring planes located near the journal bearings, at coordinates $z_2 = -21$ mm (front plane) and $z_1 = 91.5$ mm (rear plane). The center of mass is located at $z_G = 41$ mm. We set up numerical models of the rotor bearing system for a comparison with the experimental tests. We analyzed the unbalance response with a nonlinear model (orbit method) and with a linearized model, both of which make use of the stiffness and damping coefficients of the JBs. The next sub-section describes such models.

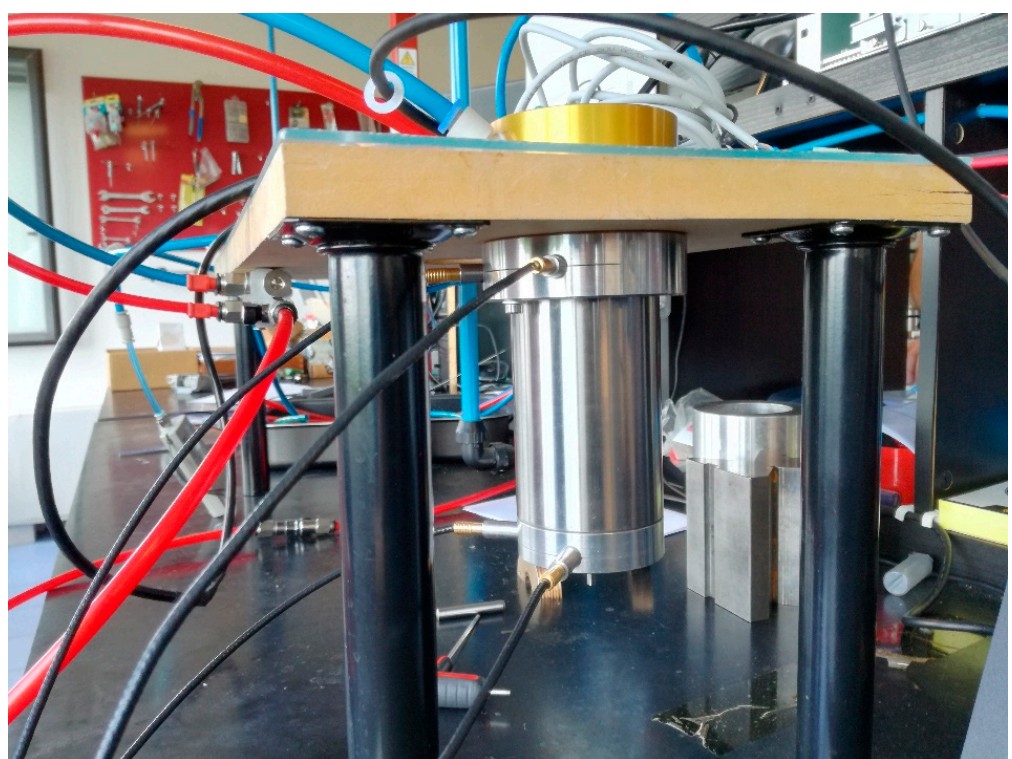

**Figure 1.** The electro-spindle under test.

**Table 1.** Geometry of the rotor bearing system.

| Variable | Description | Value |
|:---:|:---:|:---:|
| $d_s$ | Supply hole diameter | 0.15 mm |
| $h_f$ | Radial clearance on front JB | $17.5 \pm 1.5$ μm |
| $h_r$ | Radial clearance on rear JB | $21.5 \pm 1.5$ μm |
| $l_1$ | Axial distance | 70 mm |
| $l_2$ | Axial distance | 26 mm |
| $L$ | Journal bearing axial length | 30 mm |
| $r_f$ | Radius of front JB | 12.5 mm |
| $r_r$ | Radius of rear JB | 11 mm |

**Table 2.** Inertia properties of the rotor.

| Variable | Description | Value |
|:---:|:---:|:---:|
| $m$ | Mass | 382 g |
| $I$ | Transverse moment of inertia | 667 kg·mm$^2$ |
| $I_p$ | Polar moment of inertia | 31 kg·mm$^2$ |

### 2.1. Nonlinear Model

Figure 2 illustrates the spindle with two journal bearings located at distance $l_1$ and $l_2$ with respect to the center of mass. The bearings can be considered fixed as the bushings were mounted in the external carter of the electrospindle with a small interference to prevent their radial vibration. In this way, no additional external damping is introduced in the system. The amplitude of the shaft orbits resulting from a static unbalance eccentricity $e = 1$ μm is calculated solving the time-dependent Reynolds equation coupled with the 4 degrees of freedom (DOF) rotor dynamic model [15].

The 4 DOF rotodynamic equations are:

$$
\begin{aligned}
m\ddot{x}_C - F_{cx} - F_{extx} &= me\omega^2 \cos\omega t \\
I\ddot{\vartheta}_y - I_p\omega\dot{\vartheta}_x - M_{cy} + F_{extx}(z_G - z_F) &= (I - I_p)\omega^2\gamma\cos(\omega t + \varphi_1) \\
m\ddot{y}_C - F_{cy} - F_{exty} &= me\omega^2 \sin\omega t \\
I\ddot{\vartheta}_x + I_p\omega\dot{\vartheta}_y - M_{cx} - F_{exty}(z_G - z_F) &= (I_p - I)\gamma\omega^2\sin(\omega t + \varphi_1)
\end{aligned}
\tag{1}
$$

where $x_c$ and $y_c$ are the radial coordinates of the geometrical center of the rotor section with respect to its center of mass $G$, and $\vartheta_x$ and $\vartheta_y$ are the rotations around the $x$ and $y$ axes (see Figure 3). The external force applied on the rotor nose is $F_{ext}$.

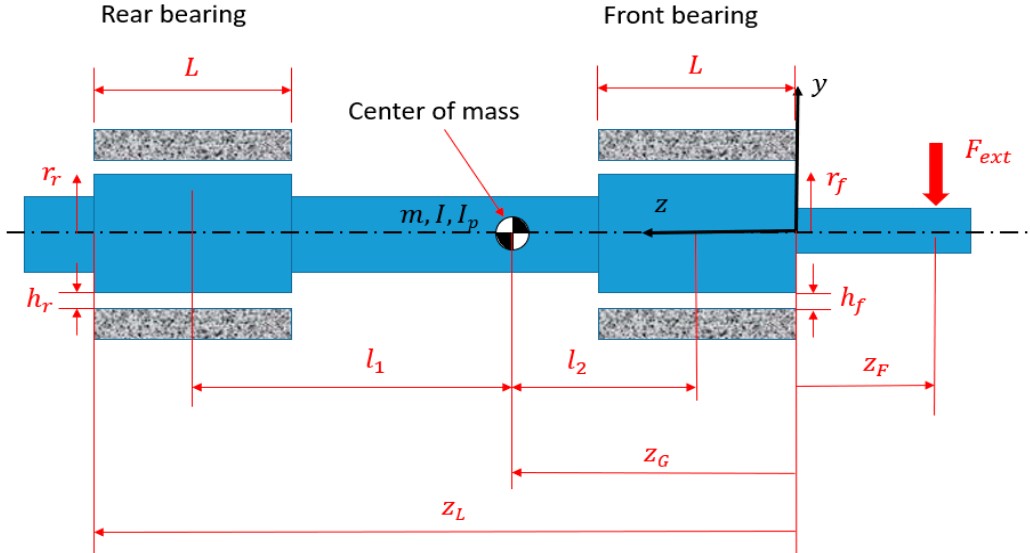

**Figure 2.** Sketch of the rotor journal bearing system.

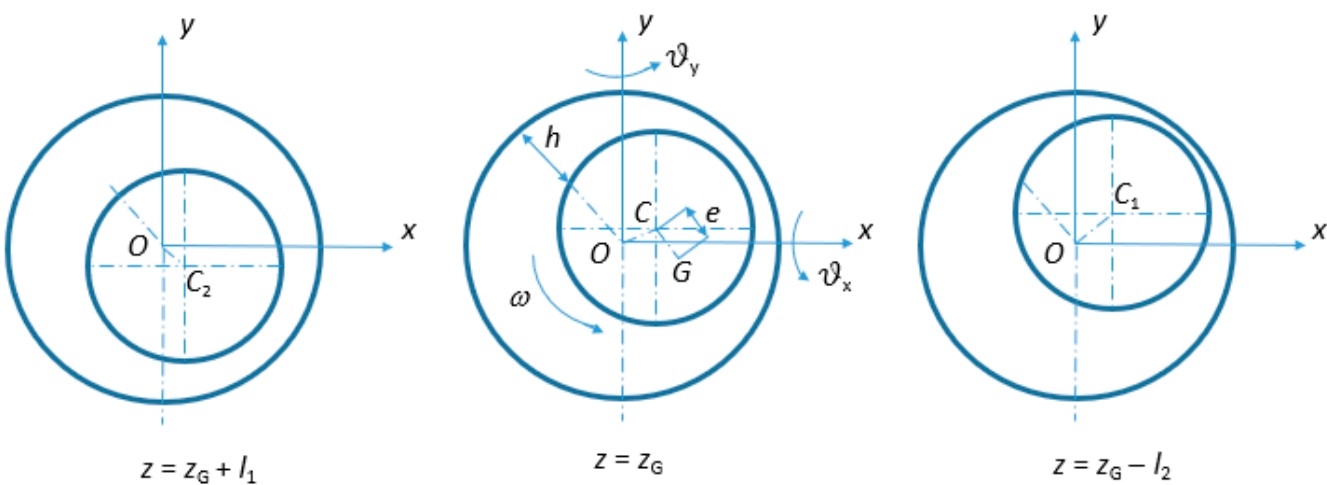

**Figure 3.** Sections of the rotor bearing system in the middle planes of journal bearings (**left** and **right**) and in the rotor center of mass (**middle**).

The reaction forces $F_c$ in bearings are given by,

$$
\begin{bmatrix} F_{cx} \\ F_{cy} \end{bmatrix} = \int_0^L \int_0^{2\pi} \left( -p \begin{bmatrix} \cos\vartheta \\ \sin\vartheta \end{bmatrix} + \tau \begin{bmatrix} \sin\vartheta \\ -\cos\vartheta \end{bmatrix} \right) r\,d\vartheta\,dz
\tag{2}
$$

while the moments $M_c$ calculated with respect to the center of mass $G$ are,

$$\begin{bmatrix} M_{cx} \\ M_{cy} \end{bmatrix} = \int_0^L \int_0^{2\pi} \begin{bmatrix} (z - z_G)(p \sin \vartheta + \tau \cos \vartheta) \\ (z - z_G)(-p \cos \vartheta + \tau \sin \vartheta) \end{bmatrix} r \, d\vartheta \, dz \tag{3}$$

The shear stress in the film is [23],

$$\tau = \frac{h}{2} \frac{\partial p}{r \partial \vartheta} + \mu \frac{\omega r}{h} \tag{4}$$

where $\mu$ is the viscosity of the lubricant (air) and $h$ is the radial film thickness. The pressure field results from the solution of the time-dependent Reynolds equation, performed by the Euler first-order method [16,17].

The algorithm performed to solve the time-dependent partial differential equation (PDE) problem coupled with the ordinary differential equation (ODE) problem is known as the orbit method. This method has the advantage of taking into account the nonlinearities of bearings that cannot be neglected for eccentricity ratios higher than 0.4. On the contrary, it requires great computation efforts compared with the linearized lumped parameters method described in the Section 2.2. For a comparison of the two methods, Table 3 indicates the computing time required to perform the calculation of the unbalance response curve in the range 40–120 krpm. For the linear model, the speed range was discretized considering 1000 points, while 16 points were considered for the nonlinear model. The table indicates the time needed to simulate the two curves (see Section 4). The nonlinear model is much more time consuming as it is necessary to extinguish the transient process before converging to the limit orbit for a single rotational speed. From the comparison, the advantage of the linearized model is clear, allowing us to individuate the critical speeds in less computing time, thanks to the finer discretization.

**Table 3.** Computing times.

| Method | Time (s) |
|---|---|
| Linearized method (1000 points) | 0.5 |
| Nonlinear method (16 points) | 38,000 |

*2.2. Linearized Model*

The linearized model considers the coefficients of stiffness and damping of the air journal bearings and solves with lumped parameters the unbalance response problem. Such coefficients are computed using the nonlinear model in synchronous conditions, with $\nu = \omega$. Starting from the equations of motion (1), it is possible to express them as a function of the shaft translations $x_{c1}$, $y_{c1}$, $x_{c2}$, $y_{c2}$ in the middle planes of bearings, according to the following transformation:

$$\begin{Bmatrix} x_c \\ \vartheta_y \\ y_c \\ \vartheta_x \end{Bmatrix} = \begin{bmatrix} \frac{l_2}{l_1+l_2} & 0 & \frac{l_1}{l_1+l_2} & 0 \\ \frac{1}{l_1+l_2} & 0 & \frac{-1}{l_1+l_2} & 0 \\ 0 & \frac{l_2}{l_1+l_2} & 0 & \frac{l_1}{l_1+l_2} \\ 0 & \frac{-1}{l_1+l_2} & 0 & \frac{1}{l_1+l_2} \end{bmatrix} \begin{Bmatrix} x_{c1} \\ y_{c1} \\ x_{c2} \\ y_{c2} \end{Bmatrix} \tag{5}$$

The problem is then described in the following matrix form:

$$[M] \begin{Bmatrix} \ddot{x}_{c1} \\ \ddot{y}_{c1} \\ \ddot{x}_{c2} \\ \ddot{y}_{c2} \end{Bmatrix} + ([G] + [C]) \begin{Bmatrix} \dot{x}_{c1} \\ \dot{y}_{c1} \\ \dot{x}_{c2} \\ \dot{y}_{c2} \end{Bmatrix} + [K] \begin{Bmatrix} x_{c1} \\ y_{c1} \\ x_{c2} \\ y_{c2} \end{Bmatrix} = \{F_{dyn}\} \tag{6}$$

where,

$$[M] = \begin{bmatrix} \frac{ml_2}{l_1+l_2} & 0 & \frac{ml_1}{l_1+l_2} & 0 \\ \frac{I}{l_1+l_2} & 0 & \frac{-I}{l_1+l_2} & 0 \\ 0 & \frac{ml_2}{l_1+l_2} & 0 & \frac{ml_1}{l_1+l_2} \\ 0 & \frac{-I}{l_1+l_2} & 0 & \frac{I}{l_1+l_2} \end{bmatrix} [G] = \begin{bmatrix} 0 & 0 & 0 & 0 \\ 0 & \frac{I_p\omega}{l_1+l_2} & 0 & \frac{-I_p\omega}{l_1+l_2} \\ 0 & 0 & 0 & 0 \\ \frac{I_p\omega}{l_1+l_2} & 0 & \frac{-I_p\omega}{l_1+l_2} & 0 \end{bmatrix}$$

$$[C] = \begin{bmatrix} c_{xx,1} & c_{xy,1} & c_{xx,2} & c_{xy,2} \\ l_1c_{xx,1} & l_1c_{xy,1} & -l_2c_{xx,2} & -l_2c_{xy,2} \\ c_{yx,1} & c_{yy,1} & c_{yx,2} & c_{yy,2} \\ -l_1c_{yx,1} & -l_1c_{yy,1} & l_2c_{yx,2} & l_2c_{yy,2} \end{bmatrix}$$

$$[K] = \begin{bmatrix} k_{xx,1} & k_{xy,1} & k_{xx,2} & k_{xy,2} \\ l_1k_{xx,1} & l_1k_{xy,1} & -l_2k_{xx,2} & -l_2k_{xy,2} \\ k_{yx,1} & k_{yy,1} & k_{yx,2} & k_{yy,2} \\ -l_1k_{yx,1} & -l_1k_{yy,1} & l_2k_{yx,2} & l_2k_{yy,2} \end{bmatrix}$$

$$\{F_{dyn}\} = \omega^2 \begin{Bmatrix} me\cos\omega t \\ (I - I_p)\gamma\cos(\omega t + \varphi_1) \\ me\sin\omega t \\ (I_p - I)\gamma\sin(\omega t + \varphi_1) \end{Bmatrix}$$

The unbalance response is given by considering the real part of the complex solution,

$$\begin{Bmatrix} x_{c1} \\ y_{c1} \\ x_{c2} \\ y_{c2} \end{Bmatrix} = \Re\left[ \left([K] + ([G]+[G])j\omega - \omega^2[M]\right)^{-1}\omega^2 \begin{Bmatrix} me \\ (I-I_p)\gamma \\ -jme \\ -j(I_p-I)\gamma \end{Bmatrix} e^{j\omega t} \right] \quad (7)$$

The phases of the signals are obtained from the ratio between the imaginary and the real part:

$$\begin{Bmatrix} \varphi_{xc1} \\ \varphi_{yc1} \\ \varphi_{xc2} \\ \varphi_{yc2} \end{Bmatrix} = \tan^{-1}\left(\Im\begin{Bmatrix} x_{c1} \\ y_{c1} \\ x_{c2} \\ y_{c2} \end{Bmatrix} / \Re\begin{Bmatrix} x_{c1} \\ y_{c1} \\ x_{c2} \\ y_{c2} \end{Bmatrix}\right) \quad (8)$$

## 3. Results

In this section, the experimental waterfall tests are shown, together with the unbalance response results obtained with the nonlinear and the linearized models.

### 3.1. Experimental Tests

The experimental unbalance response was detected during a cost-down test starting from a maximum rotational speed and switching off the permanent magnet motor. This allows a rapid deceleration without induced noises on sensors. As is visible in the bottom portion of Figure 4, the signal is synchronous with the rotational speed and no extra frequencies induced by the motor are present as they are when the motor is switched on, as shown in the upper part of the figure. The amplitudes of the frequency spectra refer to the radial displacement of the rotor.

The tests were sampled at 50 kHz and data were recorded for 30 s for each test. The frequency content of signals was calculated with Fast Fourier Transform (FFT), taking a portion of data corresponding to discrete rotational speeds. Figure 5 shows the spectrograms of the signals $x_f$ and $x_r$ measured in accordance with the measuring planes near the front and rear bearings. The waterfall diagrams were thus obtained (see Figure 6). Only the synchronous component is visible, together with the 2× component. No subsynchronous whirl appears at a supply pressure of 0.7 MPa.

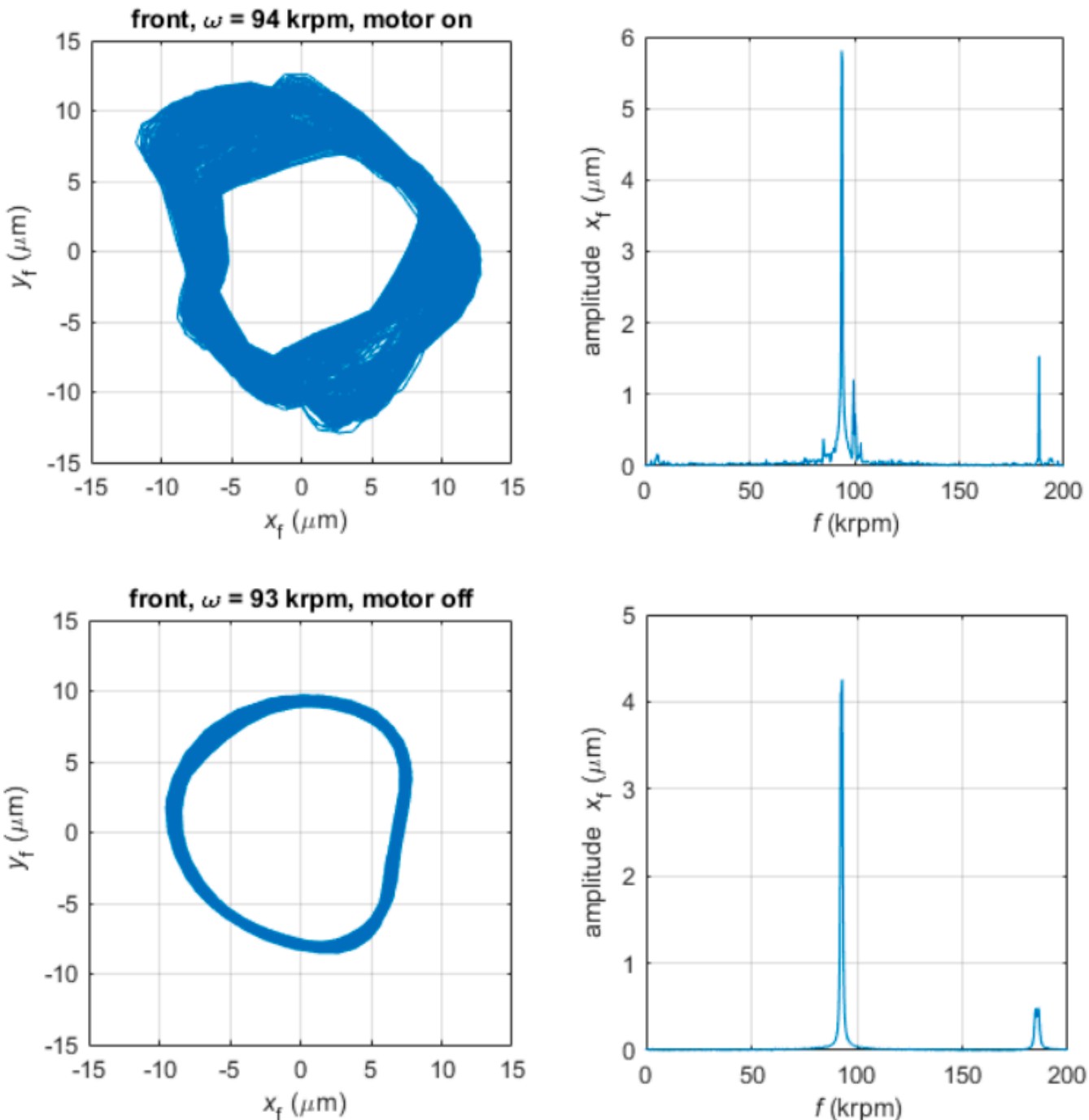

**Figure 4.** Rotor orbits and frequency spectra; comparison between active motor and disabled motor cases; $p_s$ = 0.7 MPa.

The rotor displacement in its center of mass plane and the tilting angles of rotation around axes $x$ and $y$ are calculated using Equation (5) based on the displacements measured in relation to the bearings. Figure 7 depicts the amplitude of the synchronous peak of such signals at different rotational speeds. A critical speed is evidenced at around 94 krpm. To preserve the integrity of the prototype, the critical speed was not overcome as the amplitude of the vibration increased rapidly around 94 krpm.

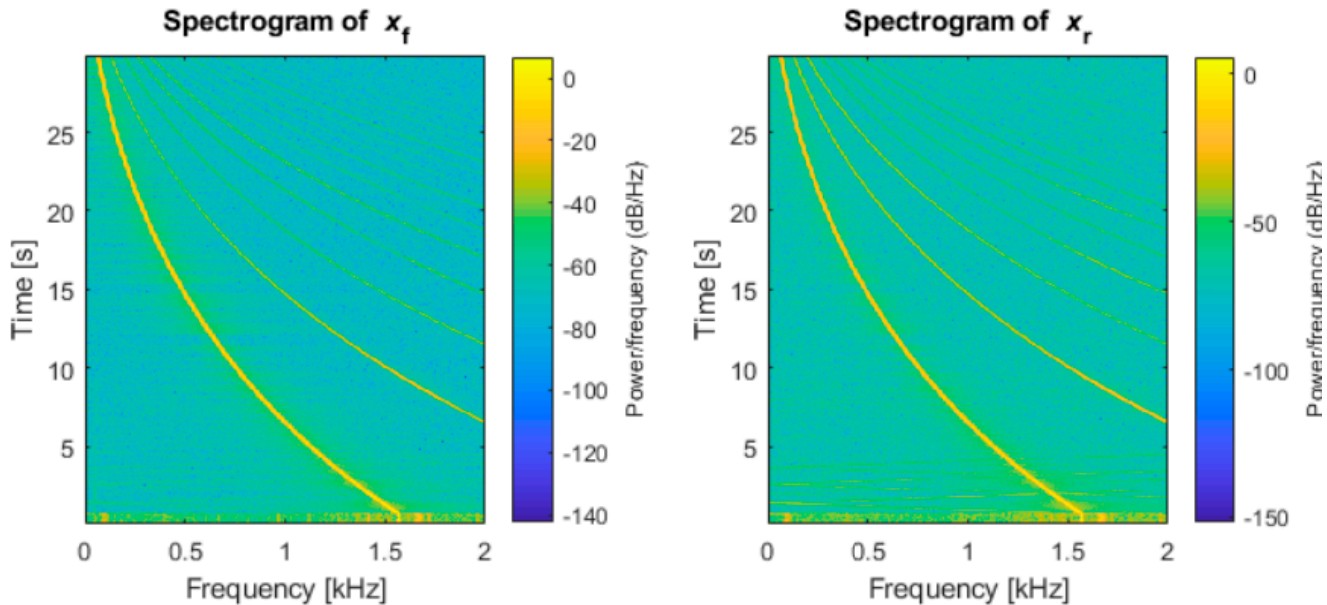

**Figure 5.** Spectrograms of signals on front and rear bearings during a deceleration test lasting 30 s and starting from 94 krpm.

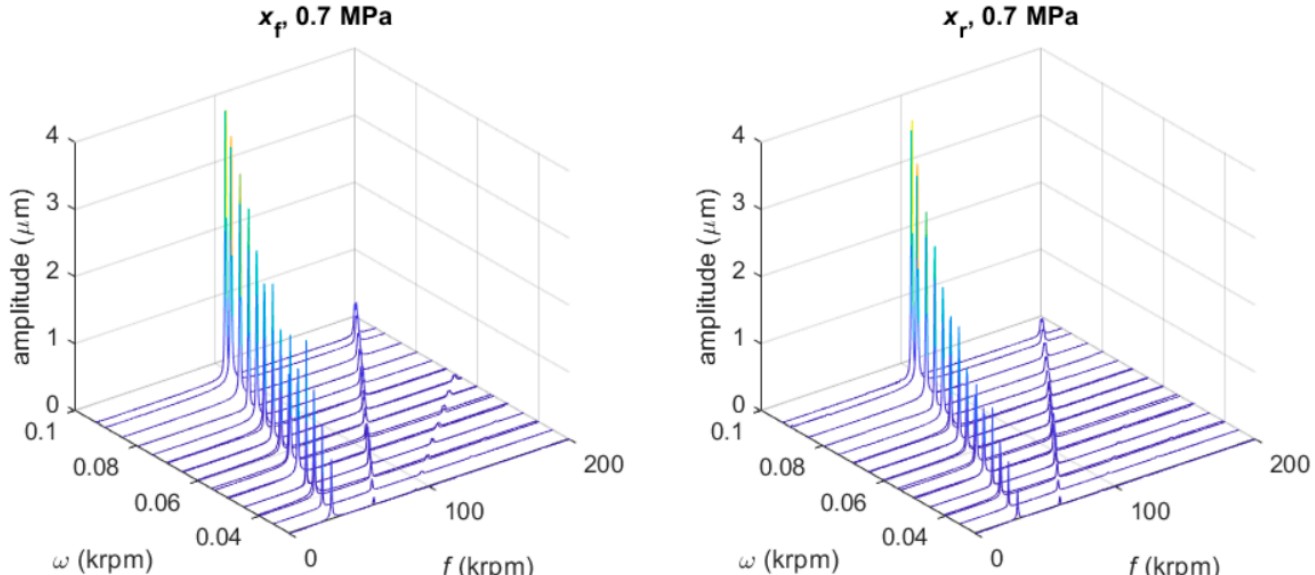

**Figure 6.** Waterfall diagrams of signals on front and rear bearings during a deceleration test lasting 30 s and starting from 94 krpm; $p_s$ = 0.7 MPa (absolute).

To further investigate the onset of instability, the supply pressure was decreased during rotation at 20, 30 and 40 krpm until a self-excited whirl was detected. An example of the frequency spectrum of the signal measured at 40 krpm is given in Figure 8. The pressure values (absolute) found for the stability threshold at 20, 30 and 40 krpm are 0.14, 0.17 and 0.19 MPa, respectively.

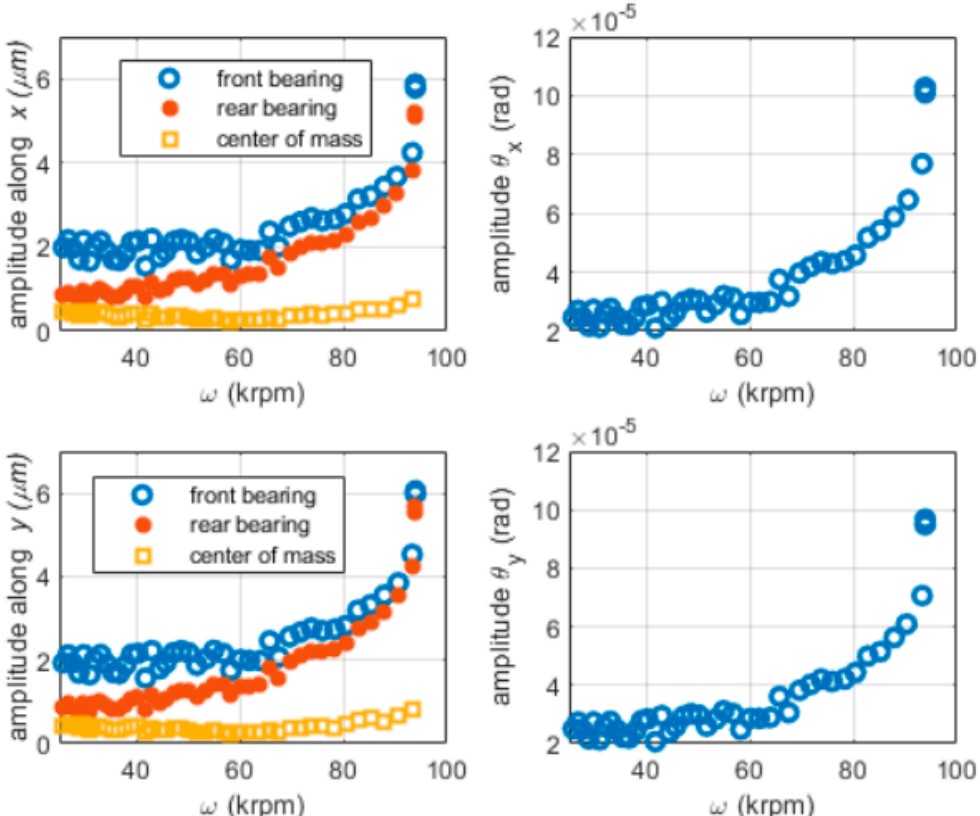

**Figure 7.** Amplitude of the synchronous component of the signals measured on front and rear bearings and of the calculated signals (center of mass and tilting angles); $p_s$ = 0.7 MPa.

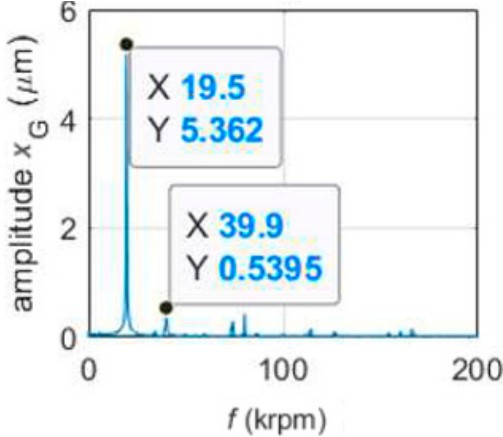

**Figure 8.** Frequency spectrum of the signal measured at 40 krpm at the stability threshold.

### 3.2. Nonlinear Model

The nonlinear model was used to simulate the unbalance response at discrete rotational speeds, with supply pressures equal to 0.7 and 0.5 MPa (absolute). The dynamic unbalance in calculations was considered null ($\gamma = 0$), while the static unbalance eccentricity $e$ was considered equal to 1 μm. Calculations were performed considering two values of air gaps for each journal bearing to take into account the manufacturing tolerances. In this way, the sensitivity of the film thickness in the unbalance response was numerically evaluated. A sufficiently small-time step was employed in order to assure the independency of the results on the time step $dt$. Value $dt = 2 \times 10^{-8}$ s was verified to be an adequate compromise between the stability of the numerical scheme and computational effort [17]. The orbits

obtained at 90 krpm are shown in Figure 9 as an example. The initial condition in this case was expressed by:

$$x(z = 0, t = 0) = 0, \ y(z = 0, t = 0) = 0$$

$$x(z = z_L, t = 0) = 0, \ y(z = z_L, t = 0) = 0$$

$$\dot{x}(z = 0, t = 0) = 0, \ \dot{y}(z = 0, t = 0) = -0.01 \text{ m/s}$$

$$\dot{x}(z = z_L, t = 0) = 0, \ \dot{y}(z = z_L, t = 0) = 0$$

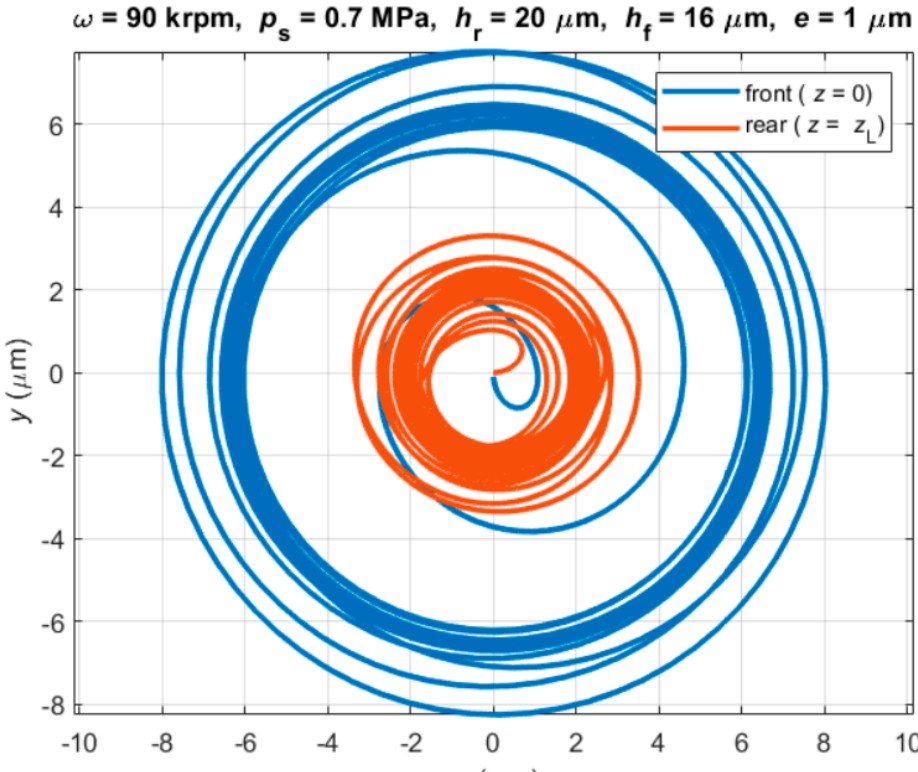

**Figure 9.** Rotor orbits, calculated with the nonlinear orbit method on front and rear bearings at $\omega$ = 90 krpm, $p_s$ = 0.7 MPa (absolute) and static unbalance eccentricity $e$ = 1 μm.

As is visible, the shaft trajectory converges to a limit cycle after an initial transient. The frequency content was evaluated calculating the FFT of the vibration signals, and the amplitude of the synchronous peak was recorded at different rotational speeds in order to reconstruct the unbalance response. The curves resulting from the nonlinear analysis are shown in Figure 10.

The onset of a subsynchronous whirl with minima air gaps and supply pressure $p_s$ = 0.7 and 0.5 MPa was detected on the front bearing for speeds above 100 krpm (see the waterfall diagrams in Figure 11). In the case of maxima air gaps with supply pressure $p_s$ = 0.7 and 0.5 MPa, a subsynchronous whirl was detected above 80 krpm (see Figure 12). The following considerations can be deduced from these numerical results:

- The front bearing is the most critical one, due to the asymmetrical JB configuration with respect to the rotor center of mass.
- Although the spindle is stable in the experimental tests at all the operating speeds, the presence of a subsynchronous whirl at high speeds in the numerical results indicates that the front bearing is near the unstable whirl onset speed when the spindle is rotated at its maximum speed (about 100 krpm).

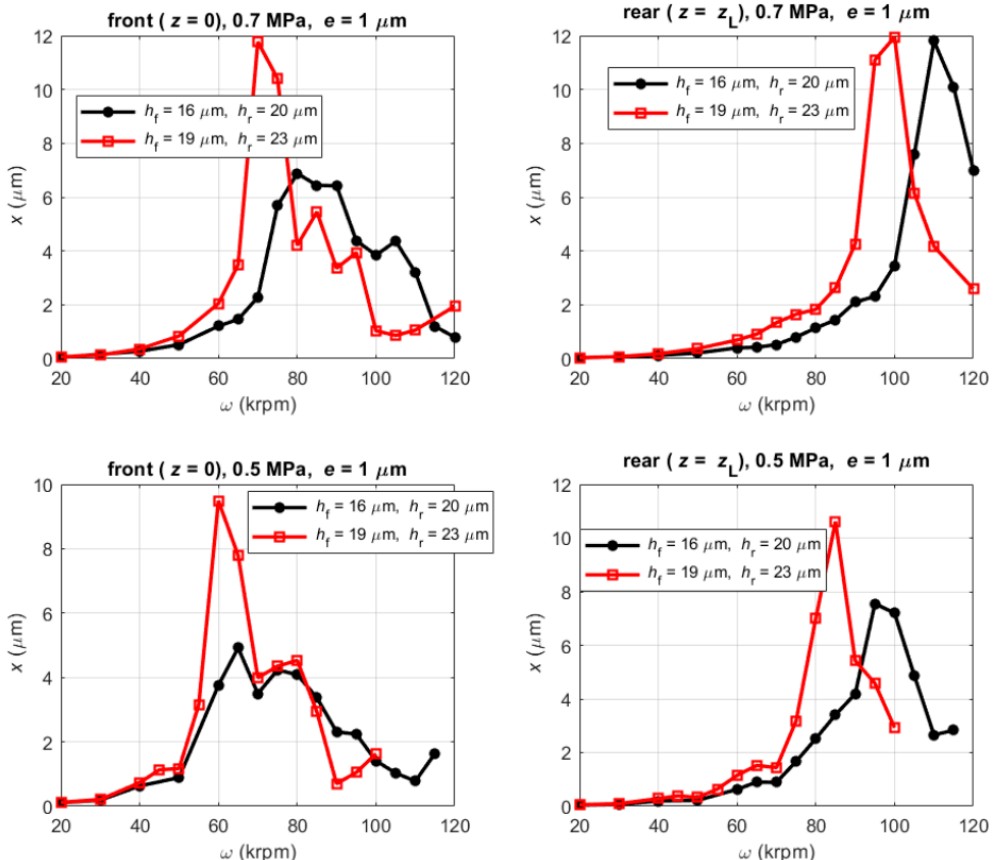

**Figure 10.** Amplitude of the synchronous component of the signals calculated with the nonlinear orbit model on front and rear bearings with $p_s$ = 0.7 and 0.5 MPa (absolute) and static unbalance eccentricity $e$ = 1 μm.

The condition of whirl instability was also investigated with the nonlinear model. The free vibration of a perfectly balanced rotor was calculated after an initial perturbation, applied to the rotor positioned in the centered condition. The damping factor of the vibration was evaluated via the logarithmic decrement method (see Ref. [22]). Figure 13 shows the damping factor of the vibration on the front bearing calculated at 20, 30 and 40 krpm at different supply pressures.

### 3.3. Linearized Model

We established a linearized model for fast calculation of the unbalance response of the rotor bearing system. This model has the advantage of providing a faster solution than the nonlinear one, and thus it is more suitable to identify the critical speeds of the system.

#### 3.3.1. Air Film Dynamic Coefficients Determination

The linear stiffness and damping coefficients of the air film were determined through the nonlinear model, as described below. A sinusoidal trajectory was imposed on the rotor and the forces transmitted to the rear and front JBs were computed.

The sinusoidal trajectory was applied along the $x$ axis:

$$x = x_0 \sin \nu t \tag{9}$$

The force components can be expressed by:

$$F_x(t) = -k_{xx}x(t) - c_{xx}\dot{x}(t); \quad F_y(t) = -k_{yx}x(t) - c_{yx}\dot{x}(t) \tag{10}$$

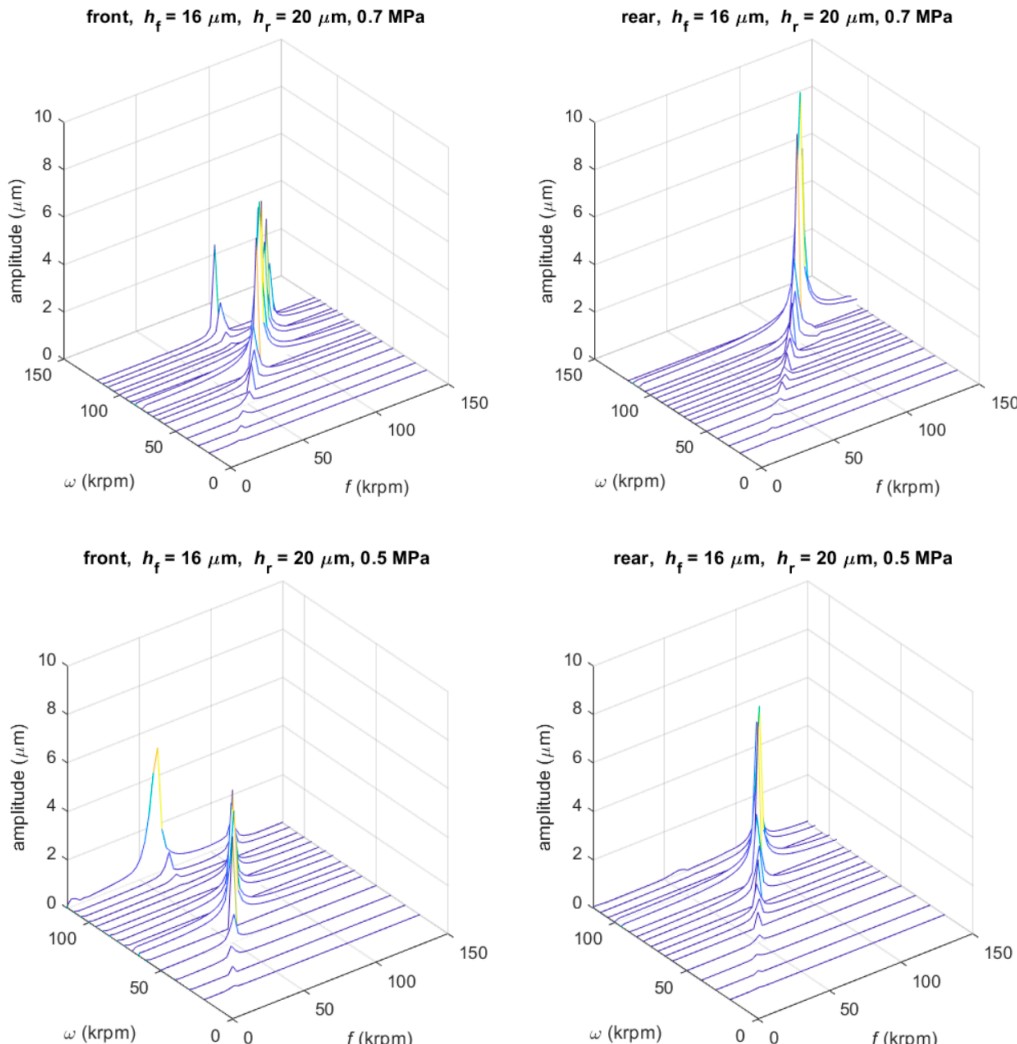

**Figure 11.** Waterfall diagram of the signals calculated with the nonlinear orbit method on front ($z = 0$) and rear ($z = z_L$) bearings with maxima film thicknesses ($h_f = 16$ μm and $h_r = 20$ μm), $p_s = 0.7$ and 0.5 MPa (absolute) and static unbalance eccentricity $e = 1$ μm.

The stiffness coefficients were computed at the time instants $t_k$ when $\dot{x} = 0$ from the calculated force components (only stiffness contribution). Similarly, the damping coefficients were computed at the time instants $t_c$ when $x = 0$ from the calculated force components (only damping contribution).

$$k_{xx} = -\left.\frac{F_x}{x}\right|_{t=t_k}; \qquad k_{yx} = -\left.\frac{F_y}{x}\right|_{t=t_k}$$
$$c_{xx} = -\left.\frac{F_x}{x_0 v}\right|_{t=t_c}; \qquad c_{yx} = -\left.\frac{F_y}{x_0 v}\right|_{t=t_c} \tag{11}$$

If calculated in the rotor centered position, the direct coefficients along x and y direction coincide, while the cross coefficients have opposite sign: $k_{yy} = k_{xx}$, $c_{yy} = c_{xx}$, $k_{xy} = -k_{yx}$, $c_{xy} = -c_{yx}$. As these coefficients are a function of both the rotational speed $\omega$ and the perturbation frequency $v$, these two parameters are, in general, independent. In case the synchronous unbalance response must be determined, it was considered $v = \omega$.

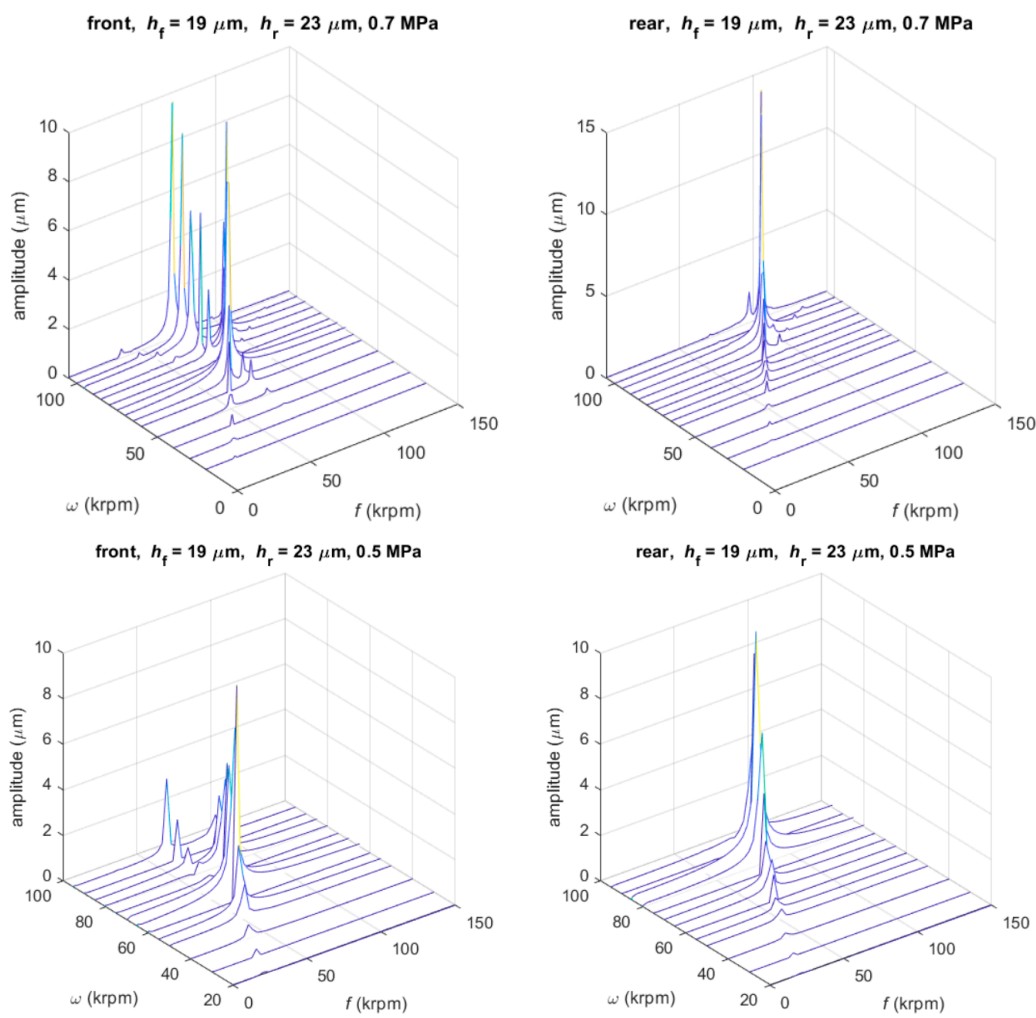

**Figure 12.** Waterfall diagram of the signals calculated with the nonlinear orbit model on front ($z = 0$) and rear ($z = z_\text{L}$) bearings with maxima film thicknesses ($h_\text{f} = 19$ μm and $h_\text{r} = 23$ μm), $p_\text{s} = 0.7$ and 0.5 MPa (absolute) and static unbalance eccentricity $e = 1$ μm.

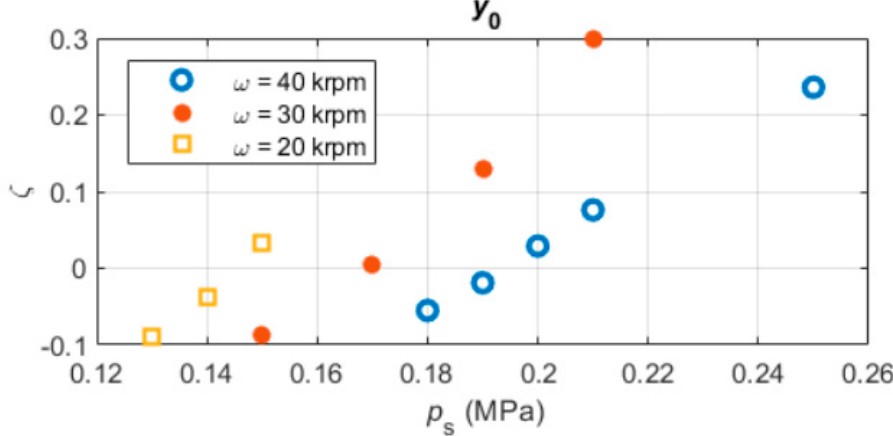

**Figure 13.** Damping factor of the signal calculated with the nonlinear orbit model on front bearing at $z = 0$ at 20, 30 and 40 krpm; $h_\text{f} = 16$ μm and $h_\text{r} = 20$ μm; static unbalance eccentricity $e = 0$ μm.

The linearized coefficients were calculated with the following conditions:

- Synchronous excitation ($\nu = \omega$);
- Small shaft eccentricity ($x_0 = 1$ μm);

- 0.5 and 0.7 MPa absolute supply pressure;
- Minima and maxima air film thicknesses (to take into account the manufacturing tolerances and the centrifugal expansion of the shaft).

The minima radial thicknesses are $h_f$ = 16 μm and $h_r$ = 20 μm on the front and the rear JBs, respectively, while the maxima values are $h_f$ = 19 μm and $h_r$ = 23 μm. The values of the coefficients are obtained at discrete perturbation frequencies and are then interpolated with polynomial expressions of second degree. The calculated and interpolated values are shown in Figure 14. The sensitivity of the coefficients with respect to the film thickness is high: an increase of 3 μm in the film thickness (18% in the case of minima film thicknesses and 15% in the case of maxima film thicknesses) has the effect of doubling the cross-stiffness and tripling the damping coefficients. The supply pressure also influences the coefficients; in particular, a decrease in the supply pressure involves an increase in direct damping coefficients and a decrease in direct stiffness coefficients. Increasing the speed in synchronous conditions, the direct stiffness increases while the direct damping decreases.

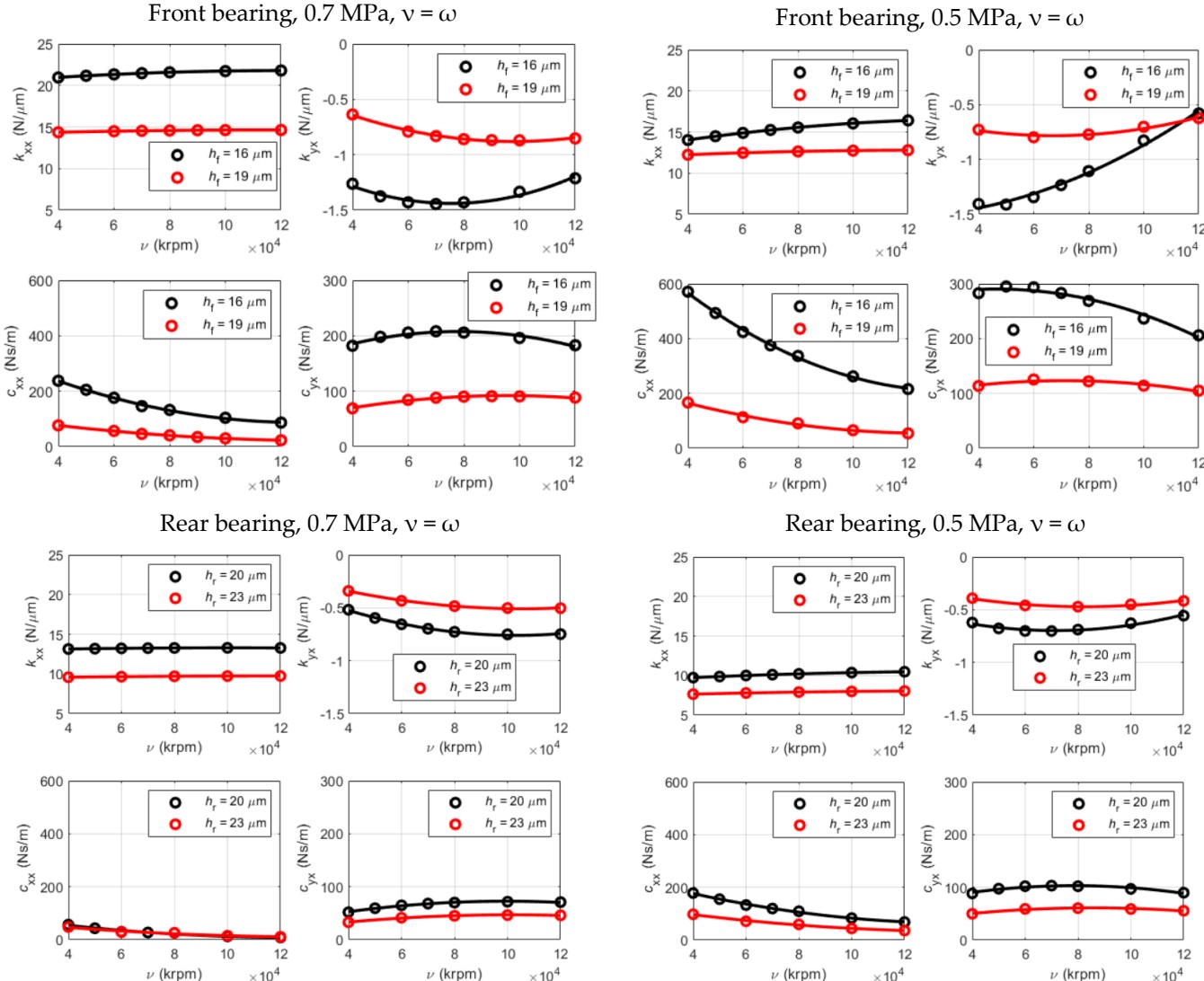

**Figure 14.** Stiffness and damping coefficients obtained in synchronous excitation conditions near the rotor's centered position; the two curves refer to minima film thicknesses ($h_f$ = 16 μm, $h_r$ = 20 μm) and maxima film thicknesses ($h_f$ = 19 μm, $h_r$ = 23 μm).

### 3.3.2. Unbalance Response

The synchronous stiffness and damping coefficients shown in Section 3.3.1 were inserted into Equation (7) to compute the linearized unbalance response. Figures 15–18 show the amplitude of the vibration measured in the front and rear bearings at 0.7 and 0.5 MPa with minimum or maximum air gaps.

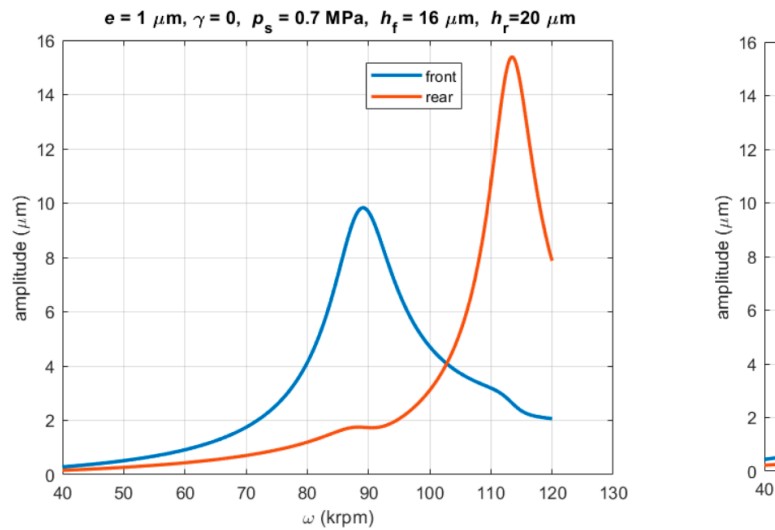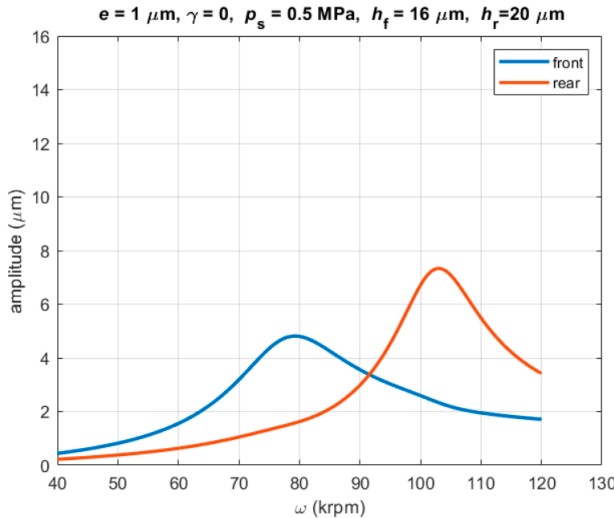

**Figure 15.** Unbalance response obtained with the linearized model; $h_f = 16$ μm, $h_r = 20$ μm, $e = 1$ μm, $\gamma = 0$, $p_s = 0.7$ and 0.5 MPa (absolute).

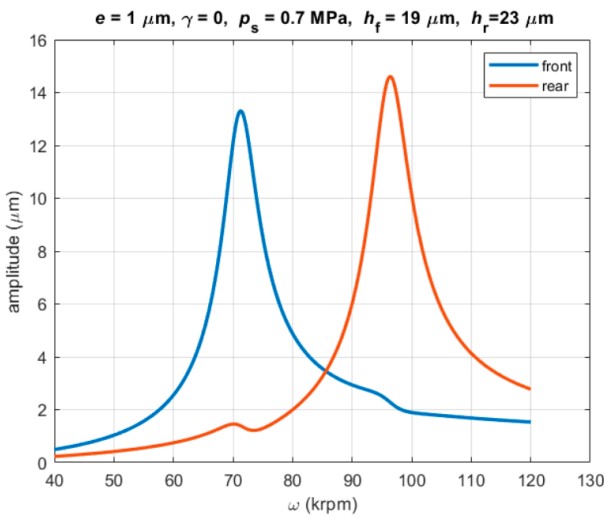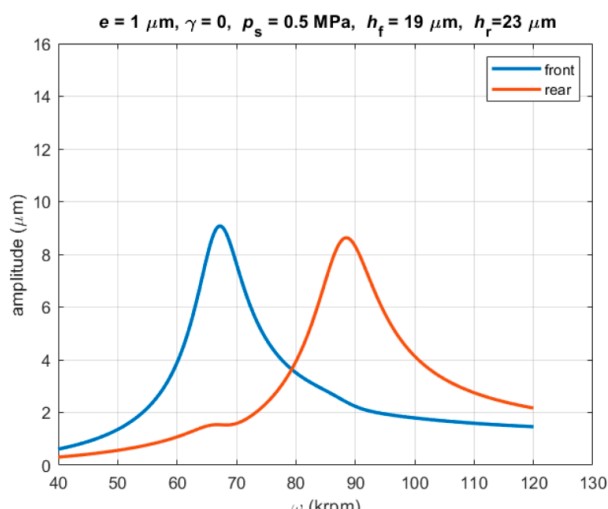

**Figure 16.** Unbalance response obtained with the linearized model; $h_f = 19$ μm, $h_r = 23$ μm, $e = 1$ μm, $\gamma = 0$, $p_s = 0.7$ and 0.5 MPa (absolute).

An increase in the film thickness of a bearing involves the decrease in the correspondent critical speed. For the geometry considered, the unbalance response curves seem to be quite independent, as the variation in the film thickness on one bearing (compare Figure 15 with Figure 17 or Figure 16 with Figure 18) has little effect on the response of the other bearing. A similar effect is visible when decreasing the supply pressure, which involves a decrease in the stiffness of all JBs. Due to the higher damping, at smaller supply pressure, the unbalance amplitude reduces.

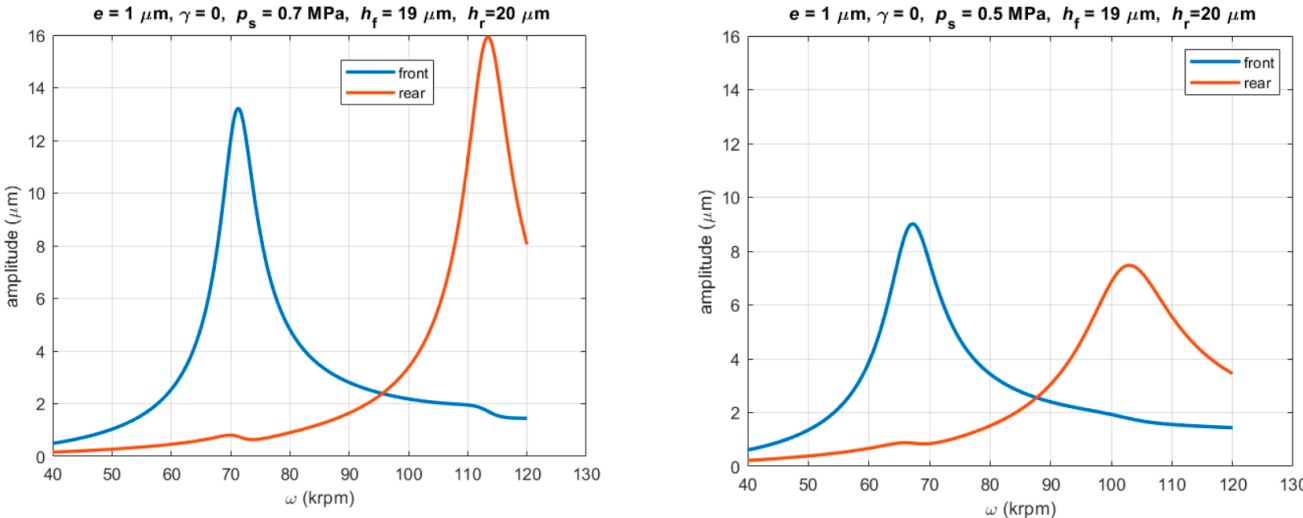

**Figure 17.** Unbalance response obtained with the linearized model; $h_f$ = 19 μm, $h_r$ = 20 μm, $e$ = 1 μm, $\gamma$ = 0, $p_s$ = 0.7 and 0.5 MPa (absolute).

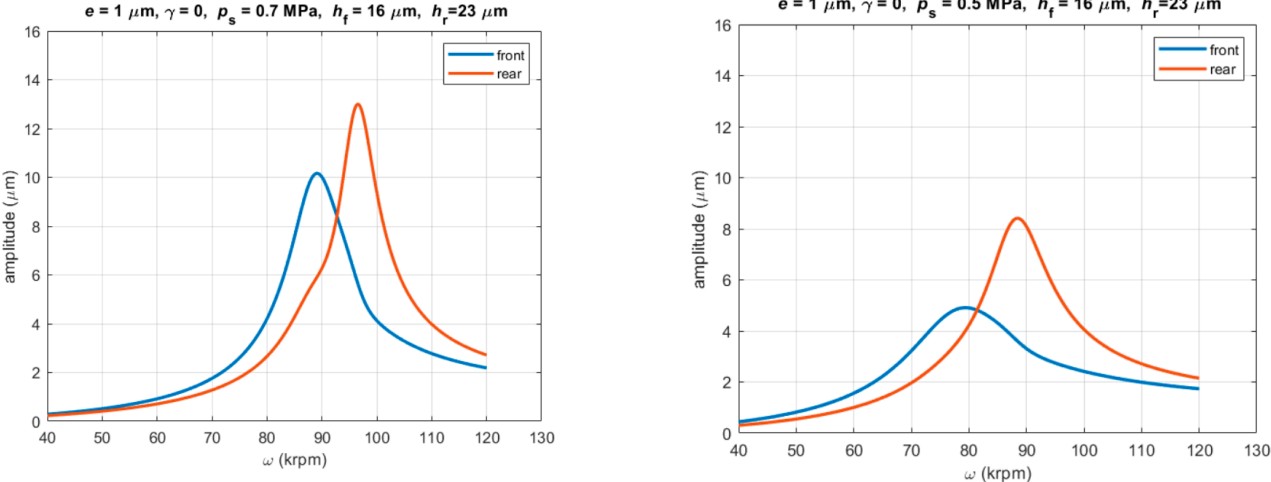

**Figure 18.** Unbalance response obtained with the linearized model; $h_f$ = 16 μm, $h_r$ = 23 μm, $e$ = 1 μm, $\gamma$ = 0, $p_s$ = 0.7 and 0.5 MPa (absolute).

The information about the phase between the signals measured on the rear bearing with respect to the front bearing is not relevant for this study. However, for the sake of completeness, both cylindrical and conical modes are involved in the two critical speeds. At the first critical speed, the phase in cases shown in Figures 15–18 is between 60° and 75°, while at the second critical speed, the phase is in the range of 70° to 110°.

## 4. Discussion

The effect of the JBs' configuration was evaluated with the linearized method calculating the unbalance response corresponding to the bearings. Different configurations were analyzed by moving the rotor center of mass in different positions. As is visible in Figure 19, we found that the amplitude of vibration on a bearing increases (or decreases) when the rotor center of mass is approached (or departed from) by the bearing. These results are useful to modify the geometry of the rotor bearing system in order to decrease the dynamic runout.

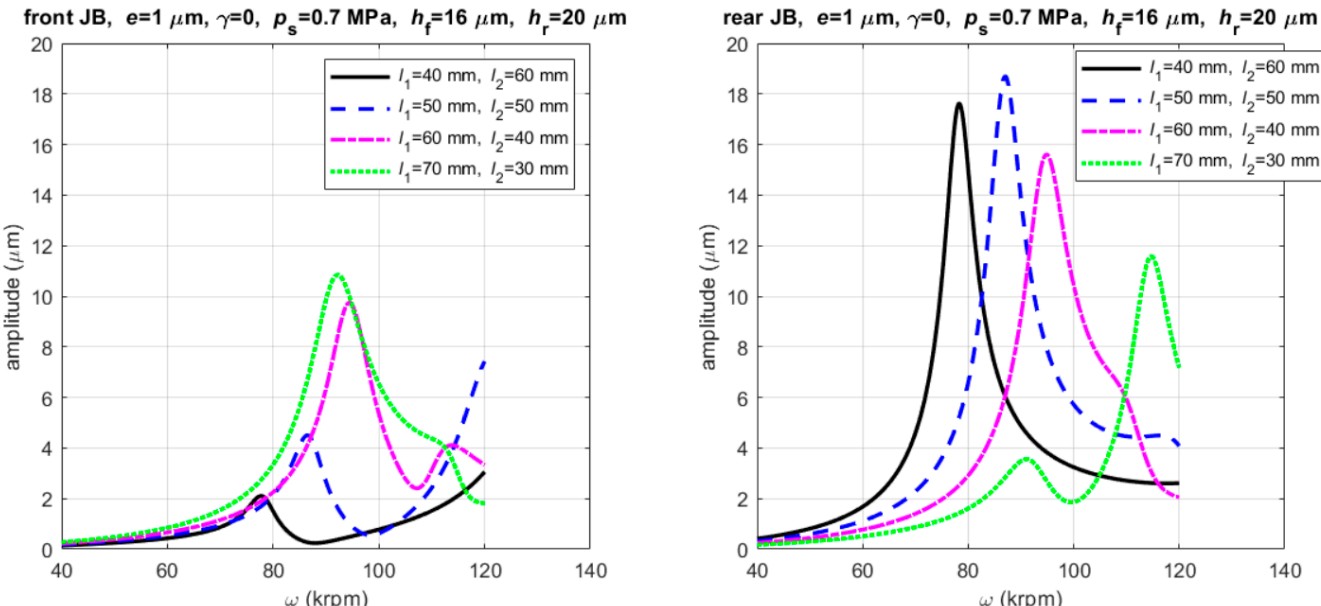

**Figure 19.** Effect of the center of mass position with respect to the JBs.

Moreover, in the case of the symmetric configuration of JBs with respect to the center of mass (same geometry of JBs and same distance $l_1 = l_2$), we verified that the two curves coincide. It is worth noting that the JBs considered in Figure 19 are the real ones, and their diameter is slightly different, so the two curves do not overlap when $l_1 = l_2$.

The unbalance responses obtained with the linearized model and with the nonlinear model are compared in Figure 20. The comparison demonstrates similar critical speeds with similar amplitudes of response. The comparison of numerical and experimental results is given in Figure 21. The numerical case that better approaches the experimental results is with $h_f = 16$ μm and $h_r = 23$ μm, which is represented in this figure. The error between the experimental first critical speed and its numerical estimation is about 10%, which is not small. Anyway, it could be accepted considering that other parameters besides the film thickness may influence the first critical speed, such as the supply orifices diameter and their discharge coefficient. Moreover, the presence of O-rings used to seal the water cooling circuit is suspected to have an influence despite the radial vibration of the bearings has been prevented with care (mounting the bushing with small interference in the external carter). The comparison was carried out essentially on the critical speeds and not on the amplitude of the displacement, as the residual unbalance of the rotor is not known. A static unbalance eccentricity $e = 1$ μm was considered in numerical simulations, together with a null dynamic unbalance. Different unbalance configurations modify the amplitude of the responses, but not the critical speeds. In the experimental tests, it was not possible to overcome the critical speed, so the curves are not complete; therefore, it was not possible to tune the static unbalance eccentricity $e$ used in the numerical models to match the dynamic runout at the critical speeds.

The values of the damping factor obtained with the nonlinear model are compared with the experimental points measured at the stability threshold. As is visible in Figure 22, the numerical model is very accurate in predicting the pressure at which the unstable whirl occurs.

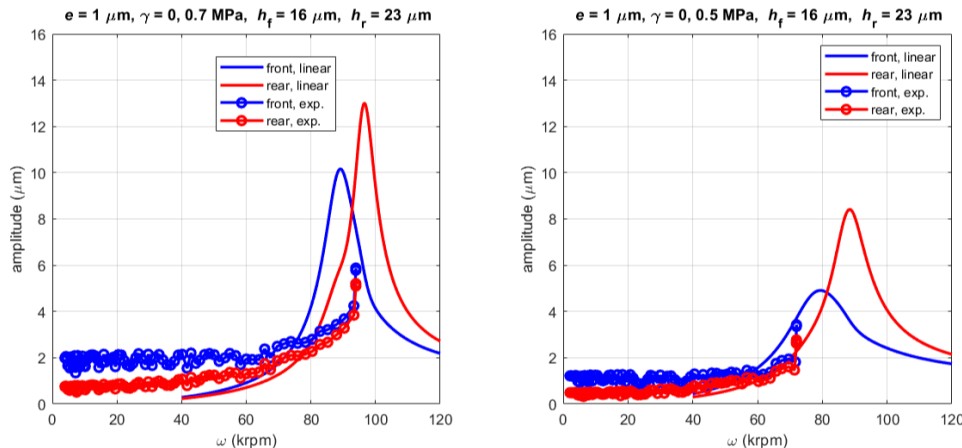

**Figure 20.** Comparison of the unbalance response obtained with the linearized model (continuous lines) and with the nonlinear model (dots); $e = 1$ μm, $\gamma = 0$, $p_s = 0.7$ and 0.5 MPa (absolute); $h_f = 16$ μm, $h_r = 20$ μm (**top**); $h_f = 19$ μm, $h_r = 23$ μm (**bottom**).

**Figure 21.** Comparison of the unbalance response obtained with the linearized model (continuous lines) and with the nonlinear model (dots); $h_f = 16$ μm, $h_r = 23$ μm, $e = 1$ μm, $\gamma = 0$, $p_s = 0.7$ (**left**) and 0.5 MPa (**right**).

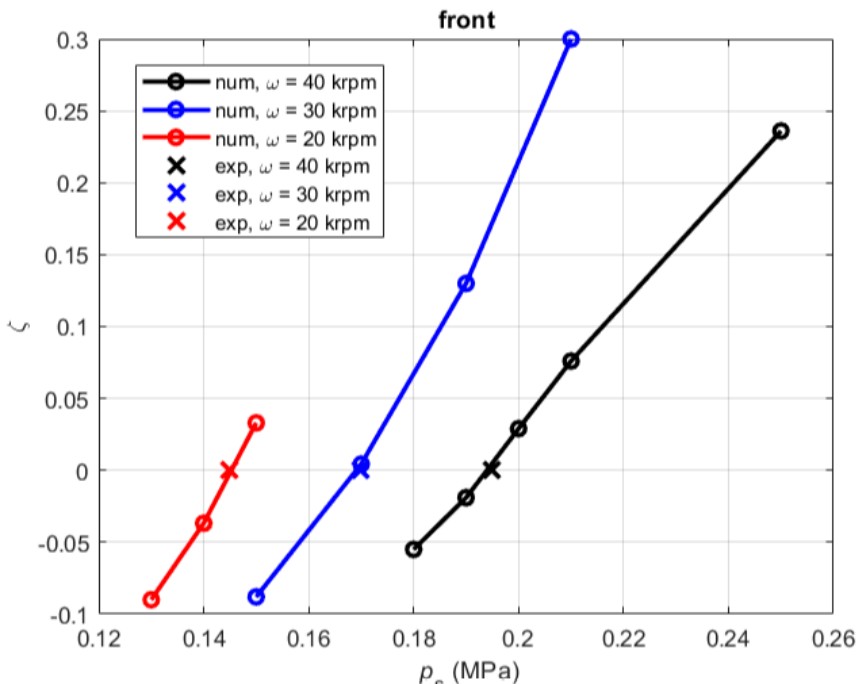

**Figure 22.** Comparison of damping factor calculated on the front bearing with the nonlinear model (continuous lines) and the experimental points at the stability threshold.

## 5. Conclusions

Based on our analysis of the unbalance response of a high-speed spindle supported by journal gas bearings, we came to the following conclusions:

- The results of the nonlinear and linearized models are in good agreement, especially regarding the critical speeds.
- The critical speeds are very sensitive with respect to film thickness; an increase in film thickness of one bearing involves a decrease in the corresponding critical speed, but has little effect on the other bearing.
- The position of the rotor center of mass strongly influences the unbalance response on bearings; in particular, the dynamic runout of the prototype measured on the front bearing can be reduced by moving the center of mass towards the rear bearing.
- The direct stiffness coefficients of the bearings decrease as the supply pressure is decreased; conversely, the direct damping coefficients increase.
- The direct stiffness coefficients of the bearings increase as the rotational speed is increased; conversely, the direct damping coefficients decrease.
- At high speeds, the front bearing is near the instability threshold because a subsynchronous frequency appears in the nonlinear model results (see Section 3.2).
- The difference between the experimental and numerical first critical speed is about 10% and it could be acceptable, considering that other parameters besides the film thickness may influence it, such as the orifices diameter and their discharge coefficient.
- The difference in the amplitude of the dynamic runout may be due to the fact that in numerical simulations, only the static unbalance eccentricity has been considered, while the experimental results can also be affected by a residual dynamic unbalance of the rotor.

Future work will focus on the stability analysis of the rotor JBs system.

**Author Contributions:** Conceptualization, F.C., L.L., A.T., T.R. and V.V.; methodology, F.C.; software, F.C.; writing—review and editing, F.C. and L.L.; validation, F.C., L.L. and A.T.; supervision, T.R. and V.V.; project administration, T.R.; funding acquisition, T.R. All authors have read and agreed to the published version of the manuscript.

**Funding:** This research was funded by Carbomech Company, Contract2021.

**Data Availability Statement:** Not applicable.

**Conflicts of Interest:** The authors declare no conflict of interest.

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
