# Peer review of "Unbalance Response Analysis of a Spindle Supported on Gas Bearings: A Comparison between Different Approaches"

_lubricants, doi:10.3390/lubricants10060127_

Round 1

Reviewer 1 Report

This is an interesting topic, however, the novelty compared to existing literature is not clear. Especially, but not only, it is questionable whether a significant novelty exists in comparison with the validated nonlinear model in [16]. Also, I find the benchmark with experimental data in Fig. 18 very poor. More experiments are needed here.

Author Response

We agree with the reviewer that it would be better to complete the experimental tests by measuring the unbalance response above the first critical speed. Unfortunately, this was not possible, as the amplitude of vibration is too high and the attempt to accelerate the rotor beyond it would cause the crash of the rotor with bearings.

Anyway, we have added in paper some tests we did to detect the onset of the so-called half speed whirl. To improve the benchmark between experimental data and numerical results, we also compared the experimental and numerical results about the onset of whirl instability at different rotational speeds.

Apart the improvements in English language (all the paper was revisited by a mother tongue), in paper we have highlighted in yellow colour the new parts and the modifications introduced.

Reviewer 2 Report

Dear Editor, Dear Author,

I read the paper titled “Unbalance response analysis of a spindle supported on hybrid bearings: a comparison between different approaches”.

The topic of this manuscript falls within the scope of Lubricants.

The paper shows a quite interesting comparison among experimental data, non-linear and linear models of a gas journal bearings. The paper is well written and structured, resulting in a clear description of the problem tackled.

I have just few comments and remarks:

  1. If possible, relevant and appropriate, add more references to balance the number of self-citations. These are ok since the authors published a lot of relevant papers on this subject.
  2. Table 1. Part of the geometry is shown in figure 2, but most of the quantities are not present. In order to clarify them, add a drawing of the bearing.
  3. Equations 1-3. The model is taken from a previous work of the authors, nevertheless few more information on the forces and quantities used in equations 1-3 should be added. A short description will help the reader.
  4. Page 4. Lines 127-128. Even if they are well-known, specify the acronym of PDE and ODE. In the page before DOF is correctly specified even it is very common acronym.
  5. Page 16, line 299. “a static unbalance of 1 m”, indeed μ prefix is missing.
  6. The authors remark a couple of times that the solving of linear model is faster than non-linear one. So, I suggest it will be interesting for the reader to know the computational time of the two approaches in that simulation.

Regards

Author Response

Apart the improvements in English language (all the paper was revisited by a mother tongue), in paper we have highlighted in yellow colour the new parts and the modifications introduced.

  1. We have improved the introduction and introduced new references to balance the self citations.
  2. We added in figure 2 the parameters that were not indicated in any scheme to help the reader. Moreover, we introduced a new figure (figure 3) to indicate the degrees of freedom and the center of the rotor in the middle of journal bearings.
  3. We added in text the explanation of viscosity and air thickness h.
  4. We have written in the extended form the meaning of acronyms PDE and ODE.
  5. µm was amended
  6. We have quantified the computing times for the linear and nonlinear methods and compared them in the new table 3

Reviewer 3 Report

A general proof reading is required, many sentences somehow read "skew" and lack clarity. The literature i most likely incomplete, reference should be made e.g. to the group of Prof. Robert Liebich at TU Berlin for gas foil bearings.

Type setting needs improvemenet e.g. in rows 66, 93, 299 Fig 9 caption

Table 1 does not match the symbols used in Fig, 2

Degress of freedom in (1) should be denoted in a figure for explanation

Matrices in equs 5, 6 and following arranged with proper column spacing

Equ. 4 \mu not introduced

Equ. 7 \cal R not described

Legend in Figs 10, 11 and more not readible

Subsection 33.2 describes observations but does not provide the interesting background knowledge to the reader

LIne 283: Waht is modiied in which way

Line 290 what is the outcome of the comparison

Last three conclusion items emain unclear lacking clarity

Line 33  what is ment with conical or cylindrical speeds?

The authors should consult a proof reader to check for a proper conclusion and to make sure the available background knowledge is transferred to the reader. Currently it is a numerical study but the hints how to improve a rotor system are not clearly stated.

Author Response

We attach a file for the response to reviewer 3.

Round 2

Reviewer 1 Report

The authors improved their manuscript which is why I recommend its publication, assuming that the following comments are addressed:

The title of the manuscript seems misleading. The authors investigate two gas bearings. The term "hybrid" does not seem appropriate, is not substantiated in the text and could therefore be deleted from the whole manuscript.

The authors state several time "static unbalance e". This is not correct. e is an eccentricity. The unbalance has the unit of kgm or gmm.

The original reference to Eq. (4) could be added (not the authors own work).

The symbols in Fig. 3 should use mathematical symbols for x,y, C2, etc. as in the equations.

Eq. 8 could be squeezed vertically by writing using the inclined divisor "/" instead of the horizontal divisor line.

What is the reason that the main spectral amplitude in Fig. 4 is only a quarter of the diameter of the orbit? The authors should check carefully whether they have accidentally divided their calculation by "2".

For better readability, it is important to scale all figures equally. For example in Fig. 7, the left figures should have the limits [0, 7] and the right figures [2, 12]. Similarly, all graphs in Fig. 11 should have the limits [0,10]. All graphs in Fig. 12 should have the limits [0,10], except the one with [0,15]. Figs. 15, 16, 17, 18, 20, 21 should have the same limit [0, 16]. Both graphs in Fig. 19 should have the same limit [0,20].

Fig. 14: all kxx should have the same limits, all kyx should have the same limits, all cxx and so on.

Are the cyy identical to the cxx in this specific setup? I miss a statement about this fact.

Fig. 9 shows the rotor orbit using the nonlinear calculation scheme. How does the result look like for the linear calculation scheme?

Eqs. (10) and (11) could be written in a single line by separating the quations with ";".

Fig. 21 is not clear. The authors write that the critical speed should be identical but the amplitudes are different due to the unknown residual unbalance. This is fine. Because of that the left graph in Fig. 21 should be deleted because the critical speed was passed and therefore the numerical settings do not match the experimental result.

Fig. 22 is not clear because the authors use the same symbols for numerical and experimental results. The symbol for experimental results needs to be different. For example, the symbol "o" could be used for numerical results (including lines connecting these points) and the symbol "square" could be used for experimental results (without a line).

Author Response

We thank the reviewer for the detailed revision. We attach our reply to each comment.

Reviewer 3 Report

Thanks for incorporating additions and changes I think, the quality has impoved and the paper can now be acceted for publication.

Author Response

Thank you.